# 3D Cell Culture as Tools to Characterize Rheumatoid Arthritis Signaling and Development of New Treatments

**DOI:** 10.3390/cells11213410

**Published:** 2022-10-28

**Authors:** Jessica Andrea Badillo-Mata, Tanya Amanda Camacho-Villegas, Pavel Hayl Lugo-Fabres

**Affiliations:** 1Unidad de Biotecnología Médica y Farmacéutica, Centro de Investigación y Asistencia en Tecnología y Diseño del Estado de Jalisco (CIATEJ), A.C. Av. Normalistas 800, Colinas de la Normal, Guadalajara 44270, Jalisco, Mexico; 2CONACYT-Unidad de Biotecnología Médica y Farmacéutica, Centro de Investigación y Asistencia en Tecnología y Diseño del Estado de Jalisco (CIATEJ), A.C. Av. Normalistas 800, Colinas de la Normal, Guadalajara 44270, Jalisco, Mexico

**Keywords:** 3D cell culture, rheumatoid arthritis, pathophysiological models

## Abstract

Rheumatoid arthritis (RA) is one of the most common autoimmune disorders affecting 0.5–1% of the population worldwide. As a disease of multifactorial etiology, its constant study has made it possible to unravel the pathophysiological processes that cause the illness. However, efficient and validated disease models are necessary to continue the search for new disease-modulating drugs. Technologies, such as 3D cell culture and organ-on-a-chip, have contributed to accelerating the prospecting of new therapeutic molecules and even helping to elucidate hitherto unknown aspects of the pathogenesis of multiple diseases. These technologies, where medicine and biotechnology converge, can be applied to understand RA. This review discusses the critical elements of RA pathophysiology and current treatment strategies. Next, we discuss 3D cell culture and apply these methodologies for rheumatological diseases and selected models for RA. Finally, we summarize the application of 3D cell culture for RA treatment.

## 1. Introduction

Rheumatoid arthritis (RA) is a chronic autoimmune disease with a higher prevalence in women than men (3:1) that affects between 0.5 and 1% of the world population [1]. The estimated prevalence of RA in Latin America ranges from 0.15% in Colombia to 2.40% in Paraguay [2,3]. In Mexico, the prevalence of RA is estimated at 1.6% in adults, and there are different ranges between regions, i.e., 0.7% in Nuevo Leon to 2.8% in the Yucatan Peninsula, possibly due to a combination of genetic and environmental factors [4,5].

RA is characterized by the accumulation of inflammatory cells in the synovial capsule, cartilage and bone erosion, and complete destruction of the cartilage in severe cases [6]. RA is a chronic condition for which is currently no effective cure, and the treatment is mainly based on reducing pain and joint inflammation. On the other hand, disease-modifying anti-rheumatic drugs (DMARDs), either synthetic or biological, delay the progression of damage, and improvements have been observed in the joints of patients [7].

The current strategies to understand the pathophysiological processes of RA and the effects of novel therapeutic agents are mainly based on preclinical in vitro trials, mostly monolayer or 2D cell culture and murine models. The 2D approach implies several challenges, mainly linked to the simplicity of the models and the lack of similar or accurate systemic responses in the murine models. For these reasons, pathophysiological models based on three-dimensional cell culture technologies and other emerging techniques, such as organ-on-a-chip, have become the most innovative option to evaluate the diverse cellular mechanisms of disease development, contributing to the evaluation of new drugs directly on human cells and the ethical principle of the 3Rs (replacement, reduction, and refinement) [8]. The 3D cell culture has been helpful for investigations of morphology, proliferation, response to stimuli, and drug prospecting since it allows manipulation of the environment and cell-cell/cell-extracellular matrix interactions to simulate stages in the development of the pathology providing precise data. These platforms help to model different diseases, for example, pancreatic cancer [9], chondrosarcoma [10], intestinal infection [11], COVID-19 [12], and autoimmune pathologies, such as lupus [13] and RA.

## 2. Pathophysiology and Treatment

There have been advances in understanding RA’s pathogenesis, but the autoimmune response remains elusive. Regardless, epidemiological research has succeeded in identifying both genetic and environmental factors contributing to the risk of RA, such as variants in the HLA class II [6], genes encoding proinflammatory cytokines [14], smoking, and air pollution [15]. At the initial stage of RA disease, primary autoantibodies, rheumatoid factor, and anti-citrullinated protein antibodies (ACPAs) begin to accumulate in sera years before the clinical onset of symptoms. Evidence shows that up to 75% of RA patients produce ACPAs, and their detection is the most specific diagnostic test for RA [16,17,18]. The rheumatoid joint is enriched in citrullination; to date, more than 100 citrullinated proteins have been identified in the RA joint.

In RA patients, the synovium is typically infiltrated by immune cells that produce a variety of proinflammatory cytokines facilitating inflammation and eventually leading to tissue destruction [14]. The synovial membrane is greatly expanded due to the increase and activation of macrophage-like synoviocytes (MLSs) and fibroblast-like synoviocytes (FLSs). MLSs produce a variety of proinflammatory cytokines, including IL-1, IL-6, and tumor necrosis factor (TNF-α), while FLSs express IL-6, matrix metalloproteinases (MMPs), and small-molecule mediators, such as prostaglandins and leukotrienes [19]. The aggressive and invasive phenotype of FLSs that forms the hyperplastic tissue, called pannus, contributes to cartilage damage by the attachment to the articular surface and local matrix destruction and cartilage degradation. In addition to acting as target cells in RA, evidence also implicated chondrocytes as effector cells in RA through the induction of inflammatory cytokines and chemokines, such as IL-6 and TNF-α, as well as matrix metalloproteinases (MMPs) and nitric oxide (NO). FLSs negatively affect the subchondral bone by activating and maturing bone-resorbing osteoclasts. Osteoclasts are highly responsive to autoantibodies, proinflammatory cytokines, and, more importantly, receptor activators of nuclear factor kappa B ligand (RANKL), which is the crucial regulator of osteoclastogenesis, all cellular inflammatory processes resume in Figure 1 [20,21].

Since RA is an inflammatory disease, the treatment includes NSAIDs (Non-Steroidal Anti-Inflammatory Drugs) and corticosteroids to reduce pain and joint inflammation as first-line therapeutic agents [22] but neither has any effect on disease activity. Early management with synthetic and biological DMARDs can slow disease progression and improve outcomes. DMARDs slow or stop inflammation by suppressing the overactive immune system. They help to modify or change the course of the disease and could even result in its remission, which is the goal of all conventional treatments [23]. Conventional synthetic DMARDs (csDMARDs) are the first-line treatment for RA, including methotrexate, sulfasalazine, and azathioprine (Table 1). Combined, they can take 6–12 months to produce symptomatic improvement [24]. For patients who fail to respond to these drugs, biologic DMARDs (bDMARDs) offer opportunities for disease management, particularly inhibitors of TNF-α [25]. Among the cytokines, TNF-α appears dominant during the inflammatory phase, promoting the activation of osteoclasts, chondrocytes, vascular endothelium, and fibroblasts while upregulating the expression of other proinflammatory cytokines [26], making it a key target. In addition, interleukin (IL) inhibitors, anti-B-cell agents, and a CD80/86 inhibitor termed cytotoxic T lymphocyte-associated protein 4-immunoglobulin (CTLA4-Ig) are also commercially available [27,28]. JAK inhibitors are the sole member of targeted synthetic DMARDs (tsDMARDs). They are low-molecular-weight compounds with oral administration that suppress the action of intracellular kinase JAK [29].

## 3. 3D Cell Culture

Monolayer cultures (2D cultures) are traditional cultures with human cells. Although they are easy to perform and reproduce, the absence of an extracellular matrix and limited cell-cell interactions result in alterations of cellular functions and rapid loss of phenotypes [35]. Additionally, there are changes in organic responses in 2D cultures, especially when there are no basolateral cell interactions such as those found in 3D cultures, mainly those coupled to synthetic or native matrices. To effectively resemble the response to drugs in 3D cell culture models derived from human cells (cell lines, mesenchymal stem cells, or primary culture), research has focused on the development of strategies such as the generation of organoids, which are three-dimensional cell culture models that self-organize in complex tissues [36], similar to organs and spheroids, cell aggregates that self-assemble in an environment that prevents attachment to flat surfaces [35]. Techniques currently used are described below. 

Cell cultures can be scaffold-based; cells are seeded on hydrogel that simulates the extracellular matrix and self-assemble into 3D spheroids [37]. This technique takes advantage of the fact that cell-matrix interactions drive cell organization. Furthermore, polymers, such as polylactic acid [38], can be used as support to replicate in vivo the extracellular matrix [39] since cells can assemble and form aggregates. The scaffold-based methodology allowed the development of a human 3D infection model to study host-pathogen interactions [11]. The dynamic culture conditions enable the formation of a polarized mucosal epithelial barrier reminiscent of the 3D microarchitecture of the human small intestine. The results suggest that the human cell-based 3D tissue model is a valuable and biologically relevant tool between in vitro and in vivo infection models to study the virulence of gastrointestinal pathogens [11].

In scaffold-free techniques, cells self-assemble in an environment that prevents attachment to surfaces. Park et al. [13] found that the scaffold-free 3D platform generates more mature cardiomyocytes than the 2D platform and it could be effective in autoimmune disease modeling including lupus.

The forced floating method uses low-adhesion plates coated with an inert substrate, such as agar or poly-2-hydroxyethyl methacrylate (poly-HEMA), to prevent cell-binding to the surface of the well [40]. The hanging drop method is another simple way to obtain spheroids. A small cell suspension is aliquoted inside the lid of a culture plate, and after reversing, microgravity concentrates cells at the bottom of the drop, and the cells stay in place due to surface tension [37]. 

In agitation-based techniques, the culture vessels with a spinner or a flask rotating around a horizontal axis keep the cell suspension in motion [41]. This agitation discourages adhesion to the vessel surface while promoting cell-to-cell contact [41].

For magnetic levitation, cells internalize magnetic nanoparticles, and a magnet is placed on top of the plate lid. Cells associate into 3D cell culture and produce ECM [39], keeping cellular activity. Similar to magnetic levitation, in acoustics-based assembly, cells are levitated into a specific spatial organization using an ultrasonic resonator [13,37].

Microfluidic systems consist of devices with micro-sized culture chambers perfused with cells and culture medium from neighboring microchannels, and cells cluster around micropillars within the chamber forming aggregates. Essential advantages of microfluidic systems are controlled mixing, chemical concentration gradients, lower consumption of reagents, and control of shear stress and pressure on cells [37,41,42].

Finally, the bioprinting technique is a method in which cell layers and supporting biological materials are positioned precisely to mimic tissue or organ functions. Entire tissues can be generated by customizing the “bio-inks” used in the assembly process (Figure 2) [37,41].

However, despite the advances in 3D cell culture the main disadvantage that needs to be addressed is the lack of reproducibility and the deficient nutrient and gas delivery [39] Microfluidic approaches can improve the reproducibility, they can be used to deliver and exchange nutrients and induce mechanical cues such as shear stress, but they tend to be more expensive and difficult to develop [35].

## 4. In Vitro Models of Joint Rheumatological Diseases

A properly functioning joint maintains the balance between anabolic and catabolic processes among all the organic components. This intricate interaction between different cell types makes it challenging to model healthy and pathological stages. Various models have been used to understand better the mechanisms involved in homeostasis and joint disease [43]. Regarding conventional 2D cell culture, it has been found that gene expression profiles in normal chondrocytes and osteoarthritic chondrocytes were taken from joint replacement surgery and showed minimal differences when growing in monolayer, suggesting that the microenvironment more influences the biological profile of the cell by the disease state of the donor [8].

Regarding three-dimensional cell culture, Sun et al. [44] developed a system for modeling osteoarthritis (OA) that allowed studying the effects of inflammatory factors and chondrocyte functions using silk supports. Žigon-Branc et al. [8] tested biological drugs in a spheroid model using osteoarthritic chondrocytes and differentiated mesenchymal stem cells, quantifying gene expression related to inflammation, such as TNF-α, interleukins, and citrullinated proteins among others. Adding proinflammatory factors and drugs allowed the evaluation to obtain results extrapolated to those found in the joints of patients with RA.

To learn more about the pathogenesis of OA, Occhetta et al. [45] developed a cartilage-on-a-chip model that allowed the application of compression to resemble the mechanical factors involved in OA pathogenesis and induce the cartilage homeostasis towards catabolism and hypertrophy. Using equine cells, Rosser et al. [46] developed a microfluidic three-dimensional chondrocyte culture mimicking essential characteristics of native cartilage that respond to biochemical injury, providing new opportunities to explore OA pathophysiology in humans and other animals.

## 5. RA and 3D Models

The implementation of procedures for developing drugs and analyzing their effects on the treatment of RA has been of particular interest, mainly those focused on evaluating in vitro systems. Traditional 2D culture systems have been of utmost importance because they offer standardized and easily reproducible conditions. Still, they lack the complex network of cell-cell and cell-matrix interactions of in vivo environments. Therefore, their ability to predict the clinical response of new components is limited [36]. These problems have been avoided by using animal models, but this strategy has disadvantages, such as the lack of homology in the organic response between animal models and humans and its inherently low reproducibility. Therefore, international organizations such as EPA (Environmental Protection Agency) in the US have declared that the future of biomedical research is to reduce tests on animals [47] and, in the same orientation, the EUROoCS (European Organ on a Chip Society) proposes the advancement of these technologies as the future in research for the development of new drugs as well as elucidating the pathophysiological functions of multiple diseases [48].

Different stages of RA development can be modeled with three-dimensional cell culture (Figure 3) based on the cells and stimuli used [21]. This methodology allows direct experimentation with human cells and this 3D in vitro approach includes scaffold-free culture, scaffold-based culture, and microfluidics (Table 2).

The treatment of RA has advanced significantly with numerous targeted therapies. However, we cannot yet predict which therapeutic agent will lead to the optimal response for each patient. In oncology, advances in patient-derived organoids (PDO) have accelerated the ability to examine variation in patient response, but little is known if it can be used for RA. The concept of synovial tissue PDO is quite appealing. If it can be developed from synovial biopsies, it would provide a powerful platform to test novel therapeutics directly in RA patient tissues [61]. A synovial biopsy is an invasive procedure; therefore, the results need to be relevant and informative. Thus, the choice of the joint to biopsy is crucial to avoid further damage [62]. It is also important to first overcome the problems related to 3D culture: the optimization and standardization of the cell culture processes.

## 6. Applications

### 6.1. Autologous Chondrocyte Implantation (ACI)

Besides helping describe certain aspects of the disease and analyze possible treatments, spheroids can be used to treat cartilage damage (Figure 4). In RA, cartilage is damaged by the inflammatory microenvironment rich in IL-1β, TNFα, IL-6, MMP, and NO. Furthermore, the autoimmune response increases the reactions for activated fibroblast-like synoviocytes and immune cells [63]. This exacerbated inflammation causes hyperplastic synovium and cartilage destruction that led to bone erosion. ACI using three-dimensional spheroids (chondrosphere) is a feasible method for treating chondral defects, with the advantage of a minimally invasive procedure. Cells are grown in patients’ own serum; no scaffold or matrix is necessary for applying chondrospheres during surgery, avoiding the potential side effects induced by allogenic or xenogenic substances. ACI using 3D spheroids for the treatment of articular cartilage lesions results in a good short- to mid-term improvement. The spheroids adhere and integrate into the cartilage defects and show a remarkable remodeling capacity [64,65]. This type of surgery has been described in the knee [66], hip [64], and shoulder [65] for OA and knee joints in RA [67]. Moreover, a Matrix Autologous Chondrocyte Implantation (MACI), as an ACI modification where chondrocyte growth on type I and III collagen membranes are used to repair a knee in RA; this technique produces chondrospheres that produce extracellular matrix [65,68] Three main topics limits the ACI or MACIs application, first the limited proliferation potential of autologous chondrocytes [68], second the need of a carrier (e.g., fibrin sealants) [69] that maintain size, structure and stability and sterility in ACI implantation in joint affected by RA. Three, now two surgeries are implicated in the ACI therapy. Those concerns demonstrate the urgent need for 3D models that break down these barriers making it possible for ACI and MACI techniques to be applied safely in more patients.

### 6.2. Mesenchymal Stem Cells (MSC)

Drug treatments are used to modulate the altered immune responses in RA, but chronic use of these drugs may cause adverse effects for many patients. Studies in regenerative medicine have used MSC derived from somatic tissues and embryonic stem cells to treat inflammatory and autoimmune diseases, such as RA. Clinical trial registrations in RA patients with MSC therapy have increased since 2011, and no toxicity or adverse effects have been found in any of the RA clinical trials conducted [70].

Ueyama et al. [71] used adipose-derived stem cells (ADSCs) spheroids to treat RA in a mouse model. They found that localized injection of spheroids reduced intra-articular inflammation and helped regenerate damaged cartilage. Osteoarthritis can be treated, too, as demonstrated in rhesus macaques injected with human embryonic stem cells spheroids on the knees, where the articular swelling was reduced [72].

Scaffolds give the cells a predetermined structure to grow and can be organic, inorganic, or both. Zhao et al. [73] used a combination of 3D-printed porous metal scaffolds and infliximab-based hydrogels. The metal scaffold is appropriate for bone defects, while the hydrogel was introduced for its anti-inflammatory, biocompatible and biodegradable properties. These composite scaffolds support ADSCs growth. The constructs were tested in a RA rabbit model and demonstrated down-regulation of inflammatory cytokines and improvement in cartilage and subchondral bone repair. Using gelatin-based microscopic hydrogel, Xing et al. [74] developed a construct to induce in vivo articular cartilage repair in a rabbit model. Mesenchymal stem cells (MSCs) were loaded into the hydrogels to form microscopic cell-laden units (microniches), which were induced to undergo self-assembly with a 3D-printed frame.

MSC in spheroids form a tight shell in the outer layer with high integrin expression and tight junction, and the inner core has looser cell-cell contact, which helps MSCs adopt a hypoxic environment through increment expression of HIF and anti-inflammatory genes [72]. These cultures can also tolerate hypothermic conditions, making them survive better and longer than dissociated MSCs in the inflamed and stressed environment, thus able to achieve therapeutic effects. From a clinical perspective, spheroids may have some advantages, as they promote the migration of large numbers of cells to the lesion site [71]. Stem cell-based therapies have shown dose-dependent effectiveness in previous studies. Therefore, higher numbers of stem cells delivered locally to the lesion in the form of spheroids may promote a favorable therapeutic effect, but more evidence about the safety and efficacy of MSC-spheroids is needed.

## 7. Conclusions

In RA, animal models and simple 2D cell cultures have contributed to elucidating some pathophysiological aspects over the past decades. Nevertheless, they are based on the simplification of all these processes, and in this scope resides the main weakness, the biomedical community needs to upgrade the research models to obtain more reliable results about the pathophysiology and, of course, to evaluate new therapeutic compounds in preclinical stages, the 3D cell culture technologies stand as a plausible solution. With the advent of technologies such as organoids and pathophysiological 3D models, it is necessary to generate an efficient and precise system of human cartilage in vitro to evaluate the impact of inflammatory mediators on synoviocytes, chondrocytes, and all cell types involved in the pathogenesis of RA, and matrices to emulate biomechanical aspects of the disease. The ideal model should simulate the role of the matrix in vivo, allowing the study of cell interactions with stem cells, one or more cell linage (human cell lines or MSC as well), and avoiding the differences related to the use of non-human animal models, making it possible to test drugs or bioactive components with optimistic activity against the disease.

## Figures and Tables

**Figure 1 cells-11-03410-f001:**
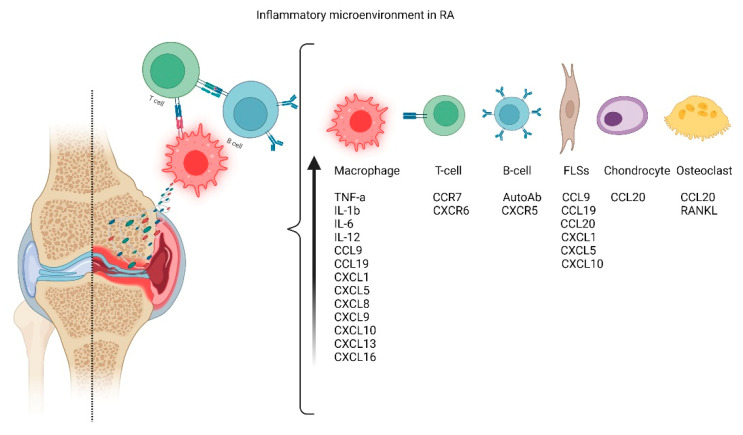
Inflammatory process. Regarding the etiology of RA, the cells responsible for articular damage include T-cells, B-Cells, Macrophages, and hyper-stimulated articular cells, such as synoviocytes (MLSs) and fibroblast-like synoviocytes (FLSs).

**Figure 2 cells-11-03410-f002:**
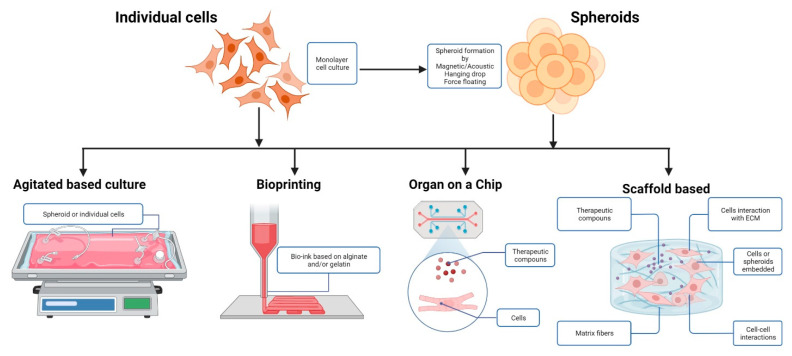
Organoid/spheroid production methods. 3D RA models, starting from 2D cell cultures, could obtain spheroids, non-adherent cell cultures, 3D bioprinting, or embedded cultures describing different aspects of the same pathology.

**Figure 3 cells-11-03410-f003:**
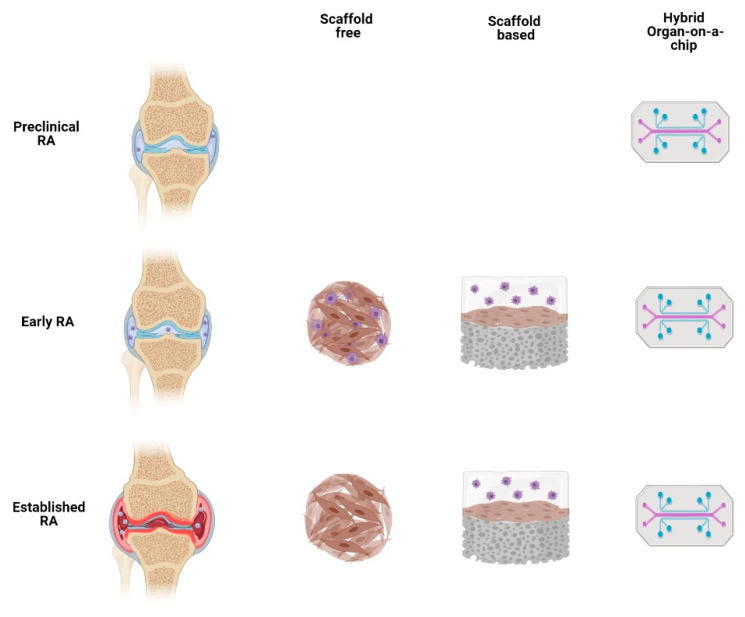
Different three-dimensional cell culture approaches can describe certain aspects of RA and even integrate themes to figure out how RA pathology affects other organs in the short and long term.

**Figure 4 cells-11-03410-f004:**
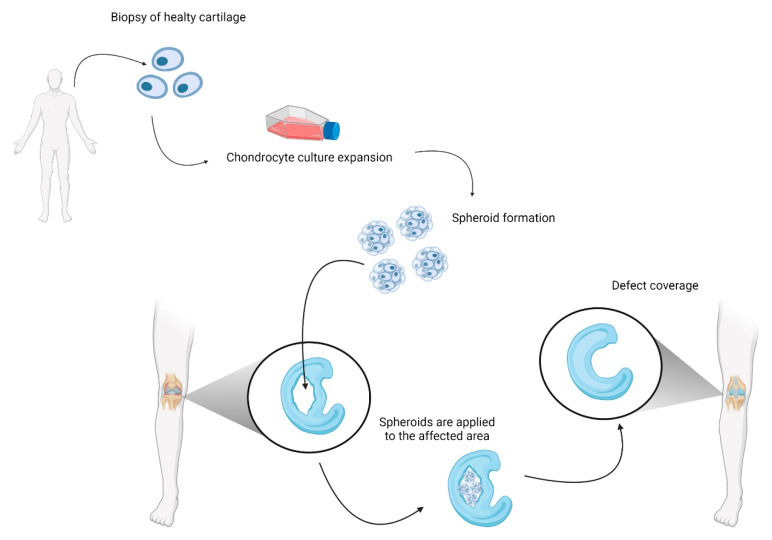
Schematic representation of autologous chondrocyte implantation (ACI) for treatment of cartilage damage in a knee with RA.

**Table 1 cells-11-03410-t001:** Overview of available treatments for RA.

Category	Example	Mechanism	FDA Approval	Side Effects	Reference
NSAID	Naproxen Ibuprofen Celecoxib	Interruption of the inflammatory cycle: blocked formation of prostaglandins through the inhibition of COX-1/COX-2 enzymes	1900 (Aspirin) *	Gastrointestinal problems including indigestion and gastric ulcers Cardiovascular, renal, or hepatic complications	[7,28]
Corticosteroids	Dexamethasone Prednisone	Modification of inflammatory mechanisms and immune responses by the activation of the cytosolic glucocorticoid receptor	1955 (Prednisone)	Bone-thinning, diabetes, high blood pressure, weight gain, immunosuppression, and psychological effects	[7,30]
csDMARD	Methotrexate Leflunomide	Interferes with deoxyribonucleotides metabolism Impedes immune cell proliferation and promotes apoptosis of these cells	1953 (Methotrexate) **	Increased risk of developing lymphoma Decreased production of hematoblast Liver, lung, skin, and epithelial damage	[31,32]
bDMARD	Etanercept Infliximab Rituximab	Inhibition of cytokines (TNF and IL) Co-stimulation blockers by binding to CD80/CD86 Anti-B-cell-agents that cause depletion, inactivation, or prevent differentiation	1998 (Etanercept)	Increased risk of frequent and severe infections Bone marrow suppression and hepatotoxicity	[27,28]
tsDMARD	Tofacitinib Upadacitinib	Binding to the adenosine triphosphate-binding site of Janus kinase (JAK) and suppression of the enzyme activity of JAK, thereby suppressing cytokine signal transduction and cytokine action	2012 (Tofacitinib)	Neutropenia/ lymphopenia/ anemia, severe infection, malignancy, major adverse cardiovascular events, and venous thromboembolism	[29]

* Not the year of approval, in 1900 aspirin was introduced to the market in the form of tablets [33]. ** MTX was not the first sDMARD approved, but it is the initial second-line drug and is considered the gold standard for RA treatment [34].

**Table 2 cells-11-03410-t002:** Selected three-dimensional models for rheumatoid arthritis.

Model	Cells	Applications	Limitations
Spheroids	Fibroblasts [49]	Analysis of phenotypic characteristics of normal and hyperplastic synovium	Simplistic The stiffness and absorption rate of these natural matrices cannot be adjusted Lack of fluid flow perfusion Accumulation of metabolites Low reproducibility
	Fibroblasts [50]	Determine effects of proinflammatory cytokines	
	Fibroblasts from patients with RA Monocytes CD14+ [51]	Analysis of hyperplasia Alteration of phenotype in macrophages Determination of the effects of proinflammatory cytokines	
	Primary chondrocytes Differentiated stem cells [8]	Test of biological anti-inflammatory drugs	
	Fibroblasts Endothelial cells [52]	Synovial angiogenesis Effect of NF-kB signaling Test of inhibitors of signaling pathways	
	Primary synoviocytes Peripheral blood mononuclear cell (PBMC) Mesenchymal stromal cell (MSC) [53]	Formation of de novo vascular structures in the context of inflammation	
Scaffold	Chondrocytes Fibroblasts [54]	Destruction of cartilage Gene expression To determine the role of genes in the pathogenesis	Reductionist No compound and oxygen gradients Lack of mechanical stimuli, such as tension and compression
	Chondrocytes Fibroblasts [55]	Pannus model Investigation of pathogenesis High-throughput drug screening	
	Macrophages Primary chondrocytes Fibroblasts [56]	Simulate pathological characteristics of cartilage with RA Determine alterations in chondrocyte phenotype Test drugs	
	Synovial fibroblasts Vascular endothelial cells [57]	Pannus model applying 3D printing technique Drug testing	
Microfluidics	Primary synoviocytes [1]	Monitor the onset and progression of synovial inflammatory responses	Challenging Difficult to operate, control, standardize and scale Difficult to adapt to high throughput screening Lack of biomechanical stimulation
	Fibroblasts Osteoclasts [58]	Predict fibroblast migration to bone cells Test drugs	
	Monocytes Primary chondrocytes [59]	Representation of healthy and disease scenario Test the therapeutic efficacy of possible treatments	
	Primary synoviocytes Primary chondrocytes [60]	Joint-on-a-chip Simulation of crosstalk between synovial and chondral organoids	

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
