# Peer review of "3D Cell Culture as Tools to Characterize Rheumatoid Arthritis Signaling and Development of New Treatments"

_cells, 2022, doi:10.3390/cells11213410_

Round 1
Reviewer 1 Report
Comments:
1. In the Abstract, “organic damage” is confusing. Please rewrite the sentence or explain what kind of damage belongs to organic damage occurs during the physiopathological processes?
2. In the Abstract, remove the comma after “efficient”.
3. In the Abstract, line 7, change Rheumatoid arthritis to RA.
4. In the Abstract, RA is not interacting technologies such as 3D culture and organ-on-a-chip, these technologies should be applied on RA treatment. Please rewrite the whole sentence.
5. Introduction, paragraph 2, line 3, remove there
6. Introduction, paragraph 3, line 2, change “were” to are, to keep present tense.
7. Introduction, paragraph 3, the last line, change rheumatoid arthritis to RA.
8. In section “2. Physiopathology and treatment”
Paragraph 1, line 5, The word is origination is confusing, suggest rewriting to initiation or other words, to describe the beginning of disease development
Paragraph 1, move ref 16-18 mark to the end of sentence information is mentioned firstly “and their detection is the most specific diagnostic test for RA”.
Paragraph 2, add reference to support the first and second sentences.
Acting changes to act
Paragraph 3, the end of first sentence, remove “for treatment”
Paragraph 3, line, first treatment line changes to first-line treatment
9. In section “3. 3D cell culture”
Paragraph 1, what is human 3D models?
Paragraph 2, line 2, remove can
Paragraph 3, reference 37 and 41 move up to the end of the previous sentence. Reference should be cited at the end of first-time mentioned sentence.
10. In section “4. In vitro models of joint rheumatological diseases”
Paragraph 3, OA is abbreviation and needs to be defined. Please use the full name.
11. In section “5. Rheumatoid arthritis and 3D models”
Change the title to RA and 3D models
Add citation about EPA declares the reduction of tests on animals for biomedical research.
12. In section “6. Applications”
Section 6.2, paragraph 1, change rheumatoid arthritis to RA.
First, when author introduce technologies used in 3D cell culture, such as scaffold-based cell culture, agitation-based technologies and magnetic levitation, related experiments, method, data, results and conclusion should be involved in the manuscript. These data and results would help readers to better understand these technologies in 3D culture development. Please add more words of specific details about each technology in associated paragraph.
Secondly, major part of this manuscript should focus on the application of 3D cell culture techniques to RA, exploring the mechanisms of pathogenesis and drug effectiveness. The key take-home message for readers should be what specific issues on RA patients which 2D cultures can’t be solved but 3Ds can. The comparison between 2D and 3D in RA research is the most important information the authors need to address in the manuscript. 3D model is more closely related to real life human body than 2D, but it does not mean 3D models are better in studying RA disease. There are only three paragraphs and need to add more data, and experiment to elaborate how important and critical 3D cell culture to RA disease.
At last, are there disadvantages of 3D cell culture technologies? Author should provide more information to guide the future research to overcome such disadvantages.
Author Response
We thank the reviewers for their examinations and suggestions to improve our manuscript. In the attached file Badillo-Mata et al., please find a revised document corresponding to Manuscript ID cells-1954308 entitled "3D cell culture as tools to characterize Rheumatoid Arthritis signaling and development of new treatments¨. In this revised version, the answers to the reviewer's comments are in blue using the tracking change. The detailed point by point modifications are listed below:
- In the Abstract, “organic damage” is confusing. Please rewrite the sentence or explain what kind of damage belongs to organic damage occurs during the physiopathological processes?
Reply: Authors have considered this comment, and now the text is: “…its constant study has made it possible to unravel the pathophysiological processes that cause damage.”
- In the Abstract, remove the comma after “efficient.”
Reply: Authors have considered this comment, and now the text is: “However, efficient and validated disease models are necessary….”
- In the Abstract, line 7, change Rheumatoid arthritis to RA.
Reply: Authors have considered this comment, and now the text is: “…can be applied to understand RA.”
- In the Abstract, RA is not interacting technologies such as 3D culture and organ-on-a-chip, these technologies should be applied on RA treatment. Please rewrite the whole sentence.
Reply: Authors have considered this comment, rewrote the sentence, and now the text is: “These technologies, where medicine and biotechnology converge, can be applied to understand RA.”
- Introduction, paragraph 2, line 3, remove there
Reply: Authors have considered this comment, and now the text is: “RA is a chronic condition for which is currently no effective cure, and the treatment is mainly based on reducing pain and joint inflammation.”
- Introduction, paragraph 3, line 2, change “were” to are, to keep present tense.
Reply: Authors have considered this comment, and now the text is: “…pathophysiological processes of RA and the effects of novel therapeutic agents are mainly based on preclinical trials….”
- Introduction, paragraph 3, the last line, change rheumatoid arthritis to RA.
Reply: Authors have considered this comment, and now the text is: “…and autoimmune pathologies such as lupus [13] and RA.”
- In section “2. Physiopathology and treatment”
Paragraph 1, line 5, The word is origination is confusing, suggest rewriting it to initiation or other words to describe the beginning of disease development.
Reply: Authors have considered this comment, and now the text is: “At the initial stage of RA disease, primary autoantibodies….”
Paragraph 1, move ref 16-18 mark to the end of sentence information is mentioned firstly “and their detection is the most specific diagnostic test for RA”.
Reply: Authors have considered this comment, and now the text is: “…is the most specific diagnostic test for RA [16–18]. The rheumatoid joint is enriched in citrullination…”
Paragraph 2, add reference to support the first and second sentences.
Reply: Authors have considered this comment and added a reference: “In RA patients, the synovium is typically infiltrated by immune cells that produce a variety of proinflammatory cytokines facilitating inflammation and eventually leading to tissue destruction [14].”
Acting changes to act
Reply: Authors have considered this comment, and now the text is: “In addition to act as target cells in RA, evidence also implicated chondrocytes as effector cells…”
Paragraph 3, the end of first sentence, remove “for treatment”
Reply: Authors have considered this comment, and now the text is: “Since RA is an inflammatory disease, the treatment includes NSAIDs (Non-Steroidal Anti-Inflammatory Drugs) and corticosteroids to reduce pain and joint inflammation as first-line therapeutic agents [22], but neither has any effect on disease activity.”
Paragraph 3, line, first treatment line changes to first-line treatment
Reply: Authors have considered this comment, and now the text is: “…synthetic DMARDs (csDMARDs) are the first-line treatment for RA…”
- In section “3. 3D cell culture”
Paragraph 1, what is human 3D models?
Reply: Authors have considered this comment and rewrote the sentence: “To effectively resemble the response to drugs in 3D cell culture models derived from human cells (cell lines, mesenchymal stem cells or primary culture) …”
Paragraph 2, line 2, remove can
Reply: Authors have considered this comment, and now the text is: “Cell cultures can be scaffold-based; cells are seeded on hydrogel that simulates the extracellular matrix and self-assemble into 3D spheroids [37].”
Paragraph 3, reference 37 and 41 move up to the end of the previous sentence. Reference should be cited at the end of first-time mentioned sentence.
Reply: Authors have considered this comment, and now the text is:
“Cell cultures can be scaffold-based; cells are seeded on hydrogel that simulates the extracellular matrix and self-assemble into 3D spheroids [37].”
“In agitation-based techniques, the culture vessels with a spinner or a flask rotating around a horizontal axis keep the cell suspension in motion [41].”
- In section “4. In vitro models of joint rheumatological diseases”
Paragraph 3, OA is abbreviation and needs to be defined. Please use the full name.
Reply: Authors have considered this comment, and now the text is: “…Sun et al. [44] developed a system for modeling osteoarthritis (OA)...”
- In section “5. Rheumatoid arthritis and 3D models”
Change the title to RA and 3D models
Reply: Authors have considered this comment, and now the text is: “5. RA and 3D models.”
Add citation about EPA declares the reduction of tests on animals for biomedical research.
Reply: Authors have considered this comment and added references for EPA and EUROoCs: “Therefore, international organizations such as EPA (Environmental Protection Agency) … to reduce tests on animals [47] and, in the same orientation, the EUROoCS (European Organ on a Chip Society) … of multiple diseases [48].”
- In section “6. Applications”
Section 6.2, paragraph 1, change rheumatoid arthritis to RA.
Reply: Authors have considered this comment, and now the text is: “…to treat inflammatory and autoimmune diseases, such as RA.”
First, when author introduce technologies used in 3D cell culture, such as scaffold-based cell culture, agitation-based technologies and magnetic levitation, related experiments, method, data, results and conclusion should be involved in the manuscript. These data and results would help readers to better understand these technologies in 3D culture development. Please add more words of specific details about each technology in associated paragraph.
Reply: We sincerely appreciate the suggestions and considering this comment. The authors agree with the suggestion and add the following information:
“Scaffold-based methodology allowed the development of a human 3D infection model to study host-pathogen interactions [11]. The dynamic culture conditions enable the formation of a polarized mucosal epithelial barrier reminiscent of the 3D microarchitecture of the human small intestine. The results suggest that the human cell-based 3D tissue model is a valuable and biologically relevant tool between in-vitro and in-vivo infection models to study virulence of gastrointestinal pathogens.
In scaffold-free techniques, cells self-assemble in an environment that prevents attachment to surfaces. Park et al. [13] found that the scaffold-free 3D platform generates more mature cardiomyocytes than the 2D platform and it could be effective in auto-immune disease modeling including lupus.”
Secondly, major part of this manuscript should focus on the application of 3D cell culture techniques to RA, exploring the mechanisms of pathogenesis and drug effectiveness. The key take-home message for readers should be what specific issues on RA patients which 2D cultures can’t be solved but 3Ds can. The comparison between 2D and 3D in RA research is the most important information the authors need to address in the manuscript. 3D model is more closely related to real life human body than 2D, but it does not mean 3D models are better in studying RA disease. There are only three paragraphs and need to add more data, and experiment to elaborate how important and critical 3D cell culture to RA disease.
Reply: We sincerely appreciate the opportunity to improve this review article by this reviewer's suggestion. We added the information highlighted in blue to support the relevance of 3D cell culture to RA disease. Now the text as:
Line 139 “Scaffold-based methodology allowed the development of a human 3D infection model to study host-pathogen interactions [11]. The dynamic culture conditions enable the formation of a polarized mucosal epithelial barrier reminiscent of the 3D microarchitecture of the human small intestine. The results suggest that the human cell-based 3D tissue model is a valuable and biologically relevant tool between in-vitro and in-vivo infection models to study virulence of gastrointestinal pathogens [11]. ”
Line 147 “Park et al. [13] found that the scaffold-free 3D platform generates more mature cardiomyocytes than the 2D platform, and it could be effective in autoimmune disease modeling, including lupus. ”
Line 176 “However, despite the advances in 3D cell culture, the main disadvantage that needs to be addressed is the lack of reproducibility and the deficient nutrient and gas delivery [39] Microfluidic approaches can improve the reproducibility, they can be used to deliver and exchange nutrients and induce mechanical cues such as shear stress, but they tend to be more expensive and difficult to develop [35]. ”
Line 236 “Synovial biopsy is an invasive procedure; therefore, the results need to be relevant and informative. Thus, the choice of the joint to biopsy is crucial to avoid further damage [62]. It is also important to first overcome the problems related to 3D culture: the optimization, and standardization optimize and standardize the cell culture processes. ”
At last, are there disadvantages of 3D cell culture technologies? Author should provide more information to guide the future research to overcome such disadvantages.
Reply: We sincerely appreciate the suggestions and considered this comment. We included information and now read as follows:
Line 139 “Scaffold-based methodology allowed the development of a human 3D infection model to study host-pathogen interactions [11]. The dynamic culture conditions enable the formation of a polarized mucosal epithelial barrier reminiscent of the 3D microarchitecture of the human small intestine. The results suggest that the human cell-based 3D tissue model is a valuable and biologically relevant tool between in-vitro and in-vivo infection models to study virulence of gastrointestinal pathogens [11]. ”
Line 176 “However, despite the advances on 3D cell culture the main disadvantage that needs to be addressed is the lack of reproducibility and the deficient nutrient and gas delivery [39]. Microfluidic approaches can improve the reproducibility, they can be used to deliver and exchange nutrients and induce mechanical cues such as shear stress, but they tend to be more expensive and difficult to develop [35].”
Line 236 “Synovial biopsy is an invasive procedure; therefore, the results must be relevant and informative. Thus, choosing the joint to biopsy is crucial to avoid further damage [62]. First, it is important to overcome the problems related to 3D culture, the optimization, and standardization to optimize and standardize the cell culture processes. ”
Reviewer 2 Report
In this review, authors provide insight into the application of 3D cell culture in RA. Besides, they focus on the advantages of various 3D models in terms of pathophysiological functions studies, precision medicine, and development of novel therapies for RA. This review contains some interesting aspects but still need improvement for the publication. My specific comments are:
1. In line 203, “Different stages of RA development can be modeled with three-dimensional cell culture (Figure 3)”. Please cite the reference. What are the key factors that the 3D model needs to adjust in order to mimic different stages of RA?
2. In line 216, as the author says, the concept of synovial tissue PDO is quite appealing for the precision medicine of RA. What are the potential barriers that need to be overcome for the clinical practice of PDO in RA?
3. In line 241, Why was MSC chosen in the form of spheroids to treat RA in animal models and do MSC spheroids have any advantages over single cells?
4. In line 221, “6.1 Autologous chondrocyte implantation (ACI)” seems to be unrelated to RA treatment. The study is focused on RA, so it would be better to show the direct application in RA , otherwise, it is suggested to condense this part.
5. Please explain all the abbreviations when was first used, such as “OA” in line 175 and “EPA” in line 198.
Author Response
- In line 203, “Different stages of RA development can be modeled with three-dimensional cell culture (Figure 3)”. Please cite the reference. What are the key factors that the 3D model needs to adjust in order to mimic different stages of RA?
Reply: We appreciate the reviewer's insightful suggestion; the authors agree and now the text reads as follows:
“Different stages of RA development can be modeled with three-dimensional cell culture (Figure 3) based on the cells and stimuli used [21].”
- In line 216, as the author says, the concept of synovial tissue PDO is quite appealing for the precision medicine of RA. What are the potential barriers that need to be overcome for the clinical practice of PDO in RA?
Reply: We appreciate the reviewer's suggestion, and the authors added the following text based on the suggestion:
“Synovial biopsy is an invasive procedure; therefore, the results need to be relevant and informative. Thus, the choice of the joint to biopsy is crucial to avoid further damage [62]. It is also important to first overcome the problems related to 3D culture: the optimization, and standardization of the cell culture processes.”
- In line 241, Why was MSC chosen in the form of spheroids to treat RA in animal models and do MSC spheroids have any advantages over single cells?
Reply: We appreciate the reviewer's insightful suggestion; the authors agree with the suggestion and added the following paragraph:
“MSC in spheroids form a tight shell in the outer-layer with high integrin expression and tight junction, and the inner core has looser cell-cell contact, which helps MSCs adopt a hypoxic environment through increment expression of HIF and anti-inflammatory genes [67]. These cultures can also tolerate hypothermic conditions, making them survive better and longer than dissociated MSCs in the inflamed and stressed environment, thus able to achieve the therapeutic effects. From a clinical perspective, spheroids may have some advantages, as they promote the migration of large numbers of cells to the lesion site [66]. Stem cell-based therapies have shown dose-dependent effectiveness in previous studies. Therefore, higher numbers of stem cells delivered locally to the lesion in the form of spheroids may promote a favorable therapeutic effect.”
- In line 221, “6.1 Autologous chondrocyte implantation (ACI)” seems to be unrelated to RA treatment. The study is focused on RA, so it would be better to show the direct application in RA, otherwise, it is suggested to condense this part.
Reply: We appreciate the reviewer's insightful suggestion, and the authors agree with the suggestion and added more explanation and literature e. g. case report specific for RA. Now the text read as follow:
Line 245 “ In RA, cartilage is damaged by the inflammatory microenvironment rich of IL-1β, TNF, IL-6, MMP and nitrogen oxide (NO). Furthermore, the autoimmune response increases the reactions for activated fibroblast-like synoviocytes and immune cells [63]. This exacerbated inflammation causes hyperplastic synovium and cartilage destruction that lead to bone erosion. ”
Line 257 “ [65][61] for OA and knee joint in RA [67]. Also, a Matrix Autologous Chondrocyte Implantation (MACI), as an ACI modification were chondrocyte growth on type I and III collagen membranes are used to repair a knee in RA; this technique produces chondro-spheres that produce extracellular matrix [65,68] . Three main topics limits the ACI or MACIs application, first the limited proliferation potential of autologous chondrocytes [68], second the need of a carrier (e. g. fibrin sealants) [69] that’s maintained size, structure and stability and sterility in ACI implantation in joint affected by RA. Three, now two surgeries are implicated in the ACI therapy. Those concerns demonstrate the urgent need of 3D models that break down these barriers making it possible for ACI and MACI techniques to be applied safely in more patients. ”
- Please explain all the abbreviations when was first used, such as “OA” in line 175 and “EPA” in line 198.
Reply: Authors have considered this comment, now the text is:
“…Sun et al. [44] developed a system for modeling osteoarthritis (OA)...”
“…organizations such as EPA (Environmental Protection Agency) in the US…”
Round 2
Reviewer 2 Report
The revised version solved most of my comments. There are still some syntax issues that need to be further rectified, please check the manuscript throughout carefully and improve the language.